# Rolling Bearing Fault Diagnosis Using Multi-Sensor Data Fusion Based on 1D-CNN Model

**DOI:** 10.3390/e24050573

**Published:** 2022-04-19

**Authors:** Hongwei Wang, Wenlei Sun, Li He, Jianxing Zhou

**Affiliations:** School of Mechanical Engineering, Xinjiang University, Urumqi 830047, China; wanghongwei_xju@126.com (H.W.); xju_heli@163.com (L.H.); jianzhou82923@163.com (J.Z.)

**Keywords:** end-to-end fault diagnosis of rolling bearings, data fusion, swarm decomposition, convolutional neural network

## Abstract

To satisfy the requirements of the end-to-end fault diagnosis of rolling bearings, a hybrid model, based on optimal SWD and 1D-CNN, with the layer of multi-sensor data fusion, is proposed in this paper. Firstly, the BAS optimal algorithm is adopted to obtain the optimal parameters of SWD. After that, the raw signals from different channels of sensors are segmented and preprocessed by the optimal SWD, whose name is BAS-SWD. By which, the sensitive OCs with higher values of spectrum kurtosis are extracted from the raw signals. Subsequently, the improved 1D-CNN model based on VGG-16 is constructed, and the decomposed signals from different channels are fed into the independent convolutional blocks in the model; then, the features extracted from the input signals are fused in the fusion layer. Finally, the fused features are processed by the fully connected layers, and the probability of classification is calculated by the cross-entropy loss function. The result of comparative experiments, based on different datasets, indicates that the proposed model is accurate, effective, and has a good generalization ability.

## 1. Introduction

As the core component of rotating machines, the rolling bearing has been widely used in modern industries [1,2]. The harsh load conditions and installation errors make it vulnerable to failure [3]. Faults occurring on the rolling bearings may damage to the equipment or casualties [4]. Meanwhile, other working components in the rotating machinery may also be affected by the failure of bearings. Consequently, to reduce the downtime of the equipment, it is necessary to monitor and diagnosis the work status of the rolling bearing.

The locations of frequent failures on rolling bearing are the outer ring, inner ring and the rolling elements. The cyclic impact modulation phenomenon will appear in the acceleration signals at a certain fault frequency when one of the components fails [5]. The status monitoring of rolling bearing based on acceleration signals is still the primary means [6].

Usually, the acceleration signals collected form the bearing pedestal are non-stationary and with heavy noisy. Additionally, lots of complex modulation components are contained in the signals, which introduce additional difficulties to the analyzing procedure, even fail to classify the work status of rolling bearings. 

Since the empirical mode decomposition (EMD) algorithm is proposed by Huang [7], the signal processing algorithms based on the decomposition theory have attracted the attention of researchers. After that, the ensemble empirical mode decomposition (EEMD) algorithm was proposed by Wu [8], which tackled the modal aliasing problems by adding the white noise to original signal. Additionally, to further improve the quality of decomposition, Yeh [9] proposed the complementary ensemble empirical mode decomposition (CEEMD) algorithm by adding two opposite white noise, and the reconstruction precision was improved. On the other hand, the modified ensemble empirical mode decomposition (MEEMD) was proposed by Yu [10]; based on the control of with noise’s parameters, the reconstruction errors and modal aliasing were suppressed simultaneously. To improve the CEEMD algorithm, complete ensemble empirical mode decomposition (CEEMDAN) [11] and improved complete ensemble EMD (ICEEMDAN) [12] were proposed, and the added noise component was eliminated thoroughly, and the number of decomposed modes could be decided adaptively.

In recent years, the variational mode decomposition (VMD), proposed by Dragomiretskiy [13], has been widely used in processing the signals of rotating machines, and the satisfied robustness for non-stationary signals could achieved based on the appropriate parameters. To add adaptabilities to VMD algorithm, Zhang [14] proposed the parameter adaptive VMD, based on the grasshopper optimization algorithm (GOA); the key parameters K and α could be obtained without the heavy reliance on the experience of researchers. 

Next, Georgios [15] was inspired by the intelligence swarm algorithm and proposed the swarm decomposition (SWD) algorithm. By applying the swarm filtering (SWF) with proper parameters iteratively, the oscillatory components (OC) could be extracted from the multi-components signals.

Similar to VMD, the SWF model also needs proper parameters, Miao [16] proposed OSWD algorithm by using the whale optimization algorithm to obtain the optimal values of SWD’s parameters. The features of fault in the decomposed signals by OSWD is more clearly then original SWD. However, the calculation efficiency could be improved furtherly.

Through the above analysis, the OSWD algorithm has better self-adaptability and decomposition effect than others. When the optimization algorithm is introduced into the process of signal decomposition, although there is a certain increase of computation, the feature information of faults in the obtained OCs are more obvious. 

In order to monitor the working state of bearings more accurately, lots of sensors have been installed on the equipment, and obtained huge data quantum. However, the traditional fault diagnosis methods cannot process the signals in time. Nowadays, the data-driven fault diagnosis methods have been widely developed. The optimal support vector machine (SVM) was adopted by Wang [17] to classify the fault signals of rolling bearing, and the accuracy was improved by adopting the dimension reduction strategy. The convolutional neural network (CNN) and fuzzy C-means (FCM) model were combined by Zhang [18] to identify the vibration signals of bearings, the weak features could be learned sufficient, and the signals was classified correctly. Qin [19] proposed the improved deep forest model; the features in the raw signals of bearings ware extracted by multi-grained scanning and classified by the cascade forest. Huo [20] combined adaptive multi-scale weighted permutation entropy and SVM model to realize the fault diagnosis of bearings, the robustness of the proposed model, under various SNRs, was stable. The one-dimensional dilated convolution network was adopted by Liang [21] to classify the faults of rolling bearings, based on the residual blocks and attentions modules, the signals under different noisy and load environments were classified by the model accurately. Zhao [22] adopted the adaptive deep gated recurrent unit to learn and identify the vibration signals of bearings, and the accuracy and robustness of the adopted model were verified. Hou [23] used bidirectional LSTM (Bi-LSTM) for the prediction of residual life of bearings, compared with other existing models, the Bi-LSTM model showed better accuracy. Xu [24] proposed a hybrid model for the fault diagnosis of rolling bearings, which was based on CNN and gcForest, the raw signals were converted into images by continuous wavelet transform firstly, the features were extracted by CNN and classified by gcForest, and the performance was verified by different datasets. Hao [25] adopted 1D-CNN and long short-term memory (LSTM) to classify the faults of rolling bearing; the classification accuracy of the data under different work conditions and SNRs was fairly stable. Huang [26] realized the accurate fault diagnosis of a high-speed train bogie; based on ICEEMDAN and 1-D convolutional neural network (1D-CNN), the weak features hidden in the raw signals were enhanced by the preprocessing of ICEEMDAN and classified by the 1D-CNN model. 

Based on the above analysis, the deep learning fault diagnosis model could meet the requirements of end-to-end fault diagnosis of rolling bearings, based on the huge quantum of data. Additionally, due to the excellent feature extraction capability, the 1D-CNN model has been proven to be an optimal model for the classification of time series signals of bearings.

To improve the prediction accuracy and generalization ability of the deep learning model, the data fusion strategy was introduced by researchers. Xue [27] adopted 1D-CNN and 2D-CNN models to extract the features in the raw signals, respectively, and fused them on the outputs of the models, the obtained features were sensitive for classification. Liu [28] employed the multidimensional feature fusion strategy to improve the accuracy of GFBD model for the braking system of heavy-haul train, and the accuracy was satisfied. Shang [29] proposed a multi-scale deep feature fusion model, based on information entropy, and the auto-encoder was adopted for extraction features firstly; then, the low dimensional features were fused and fed into the DBN classifier, and the accuracy was higher than the model without feature fusion. Liu [30] fused the statistical features and recessive features from multi-source signals and obtained a fault diagnosis model for rotating machine with faster speed and higher accuracy. Wang [31] proposed a three-stage feature fusion method to fuse the features of multi-modal signals of bearings by attention-based multidimensional concatenated CNN, and the result indicated that the accuracy of classification was improved. Pan [32] used VMD and WP algorithms to decomposition the vibration signals and extracted the statistical features; then, the features were fused and optimized, and the ELM model was adopted for classification, and the proposed method was verified to be effective. Wu [33] extracted the features from time- and frequency-domain at different speeds and fused them for classification; the reliability of diagnosis was improved under various operation conditions. The features in vibration and acoustic signals were extracted by Wang [34], based on the 1D-CNN model, and fused in another 1D-CNN based model, the fused features improved the accuracy of classifier under different SNRs.

In this paper, an integrated approach, based on BAS-SWD, 1D-CNN, and data fusion, for fault diagnosis of rolling bearings is proposed. The complex, non-stationary acceleration signals collected from test rig are divided and normalized by min-max strategy firstly. Afterwards, the BAS algorithm was adopted to optimize the SWD algorithm, and the preprocessed acceleration signals are decomposed into several OCs by BAS-SWD adaptively. Finally, the 1D-CNN model with feature fusion layer is constructed for the feature extraction and classification. The comparative experiments, based on different datasets, were carried out to evaluate the proposed method finally. The contributions of this paper are as follows:(1)An improved SWD algorithm, namely BAS-SWD, is proposed. By adopting the BAS algorithm to obtain the optimal parameters of SWD, the OCs with more obvious fault information can be obtained.(2)Based on the model of VGG-16, the improved 1D-CNN model is proposed to extract the weak features in multi-sensor signals. Through the well-trained model, the features in the raw signals can be extracted, and the high accuracy of classification is verified by different datasets.(3)The feature fusion strategy is introduced, and the features extracted by different convolutional blocks in the 1D-CNN model are fused in the fusion layer. Based on feature fusion, the classification accuracy of the model is improved obviously.

The sections of this paper are arranged as follows: the basic theories are mainly elaborated in the Section 2 and Section 3. The main procedure of the proposed fault diagnosis method of rolling bearing is explained in Section 4. Then, the proposed method is validated with different datasets and comparative experiments in Section 5. Finally, the conclusions are drawn in Section 6.

## 2. Optimal Swarm Decomposition

### 2.1. The Beetle Antennae Search Optimization Algorithm

The Beetle antennae search (BAS) is a new bio-heuristic optimization algorithm based on the foraging behavior of beetle [35]. When searching for food, the beetle can choose an appropriate direction of moving by detecting the difference of food odor concentrations around its two antennas. For example, the beetle will take a step to the direction of left antenna when the concentration near left antenna is stronger than the right one and vice-versa. By repeating this simple process, the beetle will reach to the point of food effectively.

To solve an optimization problem in *n*-dimension space by the BAS algorithm, the general steps are as follows:(1)Initialize the initial position *x*_0_ and distance between two antennas of the beetle *d*_0_. Notably, *d*_0_ should be large enough to improve detection ability. Additionally, then define a vector *x* = {*x_i_*, *I* = 1, 2, ... *L*} to record beetle’s position during each iteration; *L* is the maximum number of iterations.(2)Defines the fitness function, *fitness(·)*, to represent the odor concentration of food.(3)During each iteration, the direction of beetle’s antennae is initialized as a normalized random vector using Equation (1). Then, the coordinates of beetle’s two antennas can be calculated by Equation (2).
(1)$$\vec{D}=\frac{rand\left(n,1\right)}{\left\|rand\left(n,1\right)\right\|}$$
(2)$$\left\{\begin{array}{l} x_{li}=x_{i}+0.5d_{i}\cdot \vec{D}\\ x_{ri}=x_{i}-0.5d_{i}\cdot \vec{D} \end{array}\right.$$
where *rand(·)* is the random function, $\left\|\cdot \right\|$ is the 1-norm of a vector, and *x_.i_* is the coordinate of beetle’s antenna at *i*-th iteration, *x_i_* is the position of beetle, and *d_i_* is the distance between beetle’s antennas currently.(4)Calculate the value of fitness function with *x_.i_* and let the beetle take a step to the direction of the antenna, which has a smaller fitness value. After that, the position and distance between antennas and step length of beetle are updated using Equations (3)–(5).
(3)$$x_{i+1}=x_{i}+\delta_{i}\cdot \vec{D}\cdot sgn\left(fit\left(x_{li}\right)-fit\left(x_{ri}\right)\right)$$
(4)$$d_{i+1}=0.95d_{i}+0.01$$
(5)$$\delta_{i+1}=0.95\delta_{i}$$ where *sgn*(*·*) is the sign function.(5)Check the iteration stop condition (i.e., *i* reaches the maximum number of iterations *L*) is met or not; if not, repeat step (3) to (4) until the iteration stop condition is satisfied.

### 2.2. The Original Swarm Decomposition

Inspired by the intelligent behavior of swarm in nature, Georgios proposed a novel signal processing method, i.e., swarm decomposition (SWD) [15]. The core concept of SWD is extracting the oscillatory components (OC) from the raw signal by applying the swarm filter (SWF) algorithm iteratively. The specific OC will be peeled off by parameterizing SWF model properly during each iteration of SWD. Additionally, there are only residual components that do not contain any OCs that will be left when the termination condition of iterations is satisfied.

In the model of SWF, the filtering of input signal is represented by the swarm–prey hunting, as shown in Figure 1. The input time-domain signal is modelled by the positions of prey in every time step, as is shown in Equation (6).
(6)$$P_{prey}\left[n\right]=x_{i}\left[n\right],n\in \left[1,L\right]$$
where *P_prey_*[*n*] refers the position of the prey at *n*-th step, and *x_i_*[*n*] is the discrete time domain signal, whose length is *L*.

During the pursuing of the prey, the status of each agent in the hunter swarm can be modeled by its position and velocity. Attracting by the prey, the hunter swarm will move gradually toward the prey. At the same time, the whole swarm is controlled by two types of interactions; the first one is an external force, which caused by the attraction from prey to each agent in the swarm. The other one is internal force, which is caused by the interaction between every agent. These forces are formulated in Equation (7).
(7)$$\left\{\begin{array}{l} F_{outer}\left[i,n\right]=P_{prey}\left[n\right]-P_{hunter}\left[i,n-1\right]\\ F_{inner}\left[i,n\right]=\frac{1}{M-1}\times \sum_{j=1,j\neq i}^{M}f\left(P_{hunter}\left[i,n-1\right]-P_{hunter}\left[j,n-1\right]\right)\\ f\left(d\right)=\mathrm{sgn}(d)\times \ln\left(d_{c}/\left|d\right|\right)\\ d_{c}=rms(P_{prey}[\cdot ]) \end{array}\right.$$
where *F_inter_*[*i*,*n*] and *F_outer_*[*i*,*n*] are the internal and external forces on *i*-th agent in the hunter swarm at *n*-th time step. *P_hunter_*[*i*,*n*] represents the position of *i*-th agent on *n*-th time step. Additionally, *d_c_* is the critical distance to control the distribution of the hunter swarm, which is defined as the root mean square value of the input signal, the internal force between to agents should be attractive when the distance *d* is bigger than *d_c_*, and vice versa. This is described by function *f*(.); *sgn*(.) is the sign function, and *ln*(.) denotes the natural logarithm. Additionally, *M* is the population size of the hunter swarm.

Under the impact of the two types of interactions mentioned above, on each agent in the hunter swarm, the agents update its status on every time step to follow the trajectory of the prey. The new position and velocity can be calculated by Equation (8).
(8)$$\left\{\begin{array}{l} P_{hunter}\left[i,n\right]=P_{hunter}\left[i,n-1\right]+\delta \times V_{hunter}\left[i,n\right]\\ V_{hunter}\left[i,n\right]=V_{hunter}\left[i,n-1\right]+\delta \times \left(F_{outer}\left[i,n\right]+F_{inner}\left[i,n\right]\right) \end{array}\right.$$
where *V_hunter_*[*i*,*n*] denotes the *i*-th agent’s velocity at *n*-th time step. The coefficient *δ* is the flexible parameter of the hunter swarm.

By the end, the output of SWF model, which is the center point of hunter swarm at each time step, can be obtained. To increase the flexibility, Georgios suggested that, instead of the mean value, a weighted sum of agents is used as the final output of SWF, which is shown in Equation (9). The final output (OC) of the SWF model can be obtained when the deviation between two consecutive iterations is smaller than the preset threshold *T_th_*.
(9)$$x_{filtered}\left[n\right]=w\times \sum_{i=1}^{M}P_{hunter}\left[i,n\right]$$
where *w* is the weighted coefficient.

In the initialize stage of SWF, the normalized target frequency must be chosen firstly. Additionally, it is obtained from the power spectrum density (PSD) of input signal usually. The target frequency is the frequency with the peak value in the in the smooth PSD of the input signal, and each target frequency corresponding to a specific OC is selected gradually form the residual signal, as is shown in Equation (10). To improve the efficiency of SWD, a threshold *P_ωth_* should be preset to terminate the iterations of SWD.
(10)$$\omega_{n}=\underset{\omega }{\arg\max}\left\{pwelch\left(x\right)>P\omega_{th}\right\}$$
where *ω_n_* is the normalized frequency of a specific OC, *pwelch*(*x*) is the smooth PSD of input signal, and *P_ωth_* is the threshold of smooth PSD. Additionally, this value determines the fineness of SWD algorithm to extract the OCs in the input signal.

Then, the population size of the hunter swarm *M* and flexible coefficient *δ* can be calculated using Equation (11), based on the normalized target frequency obtained above.
(11)$$\left\{\begin{array}{l} M=round\left(33.46\times \omega_{n}^{-0.735}-29.1\right)\\ \delta =\left(-1.5\right)\times \omega_{n}^{2}+3.454\times \omega_{n}^{}-0.01 \end{array}\right.$$
where *round*(.) is the rounding function.

In additional, all of the agents in the hunter swarm are considered motionless at the first time step and distributed around the position of the first value of prey uniformly, as is shown in Equation (12). According to Georgios, a better result of decomposition can be obtained for most multi-component signals when the thresholds of SWF and SWD (*T_th_*, *P_ωth_*) are selected as (0.1, 0.1) and weight coefficient (*w*) for the output of SWF is 0.005.
(12)$$\left\{\begin{array}{l} P_{hunter}[i,1]=P_{prey}\left[1\right]+d_{c}\times \left(i-M/2-1\right)\\ V_{hunter}[i,1]=0\\ i\in [1,M] \end{array}\right.$$

### 2.3. Improve of Swarm Decomposition

According to the procedure of SWD, two important parameters, such as the threshold of smooth PSD *P_ωth_* and iteration deviation *T_th_*, must be preset before decomposing a multi-component signal by SWD algorithm. The parameter *P_ωth_* is the stop condition of SWD iteration, and it also play a decisive role to the number of OCs extracted from input signal. While *T_th_* is the stop condition of SWF iteration, which determines whether the central frequency of OC is accurate. Different values of these two parameters can lead to different decompose results. Take *P_ωth_* as an example, a small value leads to an increasing in the count of OCs and many of them are meaningless. On the contrary, there will be modal mixing in the OCs, which means the central frequency of OC is inaccurate, and some important information, such as the weak impact caused by the fault, may be lost. Therefore, it will affect the analyzing of signal seriously, and can even result in a failure of fault diagnosis. How to set these values properly is an important prerequisite for decomposing the acceleration signal of mechanical system with SWD algorithm.

Georgios suggested a pair of default values (0.1, 0.1), and they will be suited for most conditions. However, it is obviously not practicable for all scenarios, especially for the acceleration signals collected from the complex working condition. It is strongly recommended to select the properly values to balance the decomposition performance and computation time. Only by this, the SWD algorithm could satisfy the demand of end-to-end fault diagnosis. 

At present, the researcher’s experience plays a decisive role in the selection of proper parameters. Meanwhile, to get such values, a large number of decomposition experiments will be carried out by the researcher; these monotonous and repetitive works will take a lot of time but are not worth it. 

To tackle this problem, the BAS-SWD algorithm is proposed, which takes the advantage of the excellent ability of BAS’s global optimization to obtain the optimal value pair in a specific range. 

In order to obtain the optimal parameters of SWD by BAS, a proper fitness function, which can evaluate the decompose result, must be constructed firstly. Researchers have proposed many models, such as the indicators of kurtosis, correlation, signal energy, and so on. However, the experiment result shows that the effects of these indicators are not good enough.

Considering that the main component in the acceleration signal of mechanical system has obvious periodicity, usually, the strength of the periodic impact is associate with the severity of fault. Additionally, the envelop demodulation analysis has been proven to be an effective method to obtain the weak feature of fault from input signal. Therefore, the mean kurtosis of envelope spectrum (KES) of OCs is used in this paper to evaluate the performance of decomposition. According to the BAS algorithm, the product of the reciprocal of mean KES and decomposition time is adopted for fitness function (Equation (13)).
(13)$$\left\{\begin{array}{l} \underset{P\omega_{th},T_{th}}{\arg\min}\left\{Weight_{t}\left(1/\bar{KES}\right)\right\}\\ \bar{KES}=mean\left\{\sum_{i}^{N}kurt\left(ES_{i}\right)\right\} \end{array}\right.$$
where *kurt*(·) is the spectrum kurtosis function, *ES_i_* represents the envelope spectrum of the OC exacted by SWD, and *Weight_t_*(·) is the weight function, based on decomposition time.

Adding the time factor into the fitness function is to consider the real-time requirement of the end-to-end fault diagnosis. In the actual evaluation process, different combination values of the same result may be obtained. In this paper, the result with short time consumption will be selected as the optimal result. Additionally, the brief procedure of BAS-SWD is shown as Algorithm 1.
**Algorithm 1.** BAS-SWDInput: multi-component signalOutput: *P_ωth_*, *T_th_* Initialize parameters of BAS: *L*, *d*_0_, *δ*_0_, *n*, *P_ωth_*, *lb*, *P_ωth_ub_*, *T_th_lb_*, *T_th_ub_* Definition of fitness function: *fit*(*·*) ← Equation (13) Initialize of variables: *x*_0_ Definition of list of best positons: *best_p* ← *Array* [] Definition of list of best values: *best_v* ← *Array* [] *i* ← 0 while *i* < *L* − 1  Calculate fitness value: *fit* ← *fit*(*x_i_*)  Save the position: *best_p*[*i*] ← *x_i_*  Save the fitness value: *best_v*[*i*] ← *fit*  Calculate the head’s toward: D⇀ ← Equation (1)  Calculate the position of antennas: x·i ← Equation (2)  Update the positon of beetle: xi ← Equation (3)  Update the position of antennas: di+1 ← Equation (4)  Update the step length: δi+1 ← Equation (5)  *i* ← *i* + 1 end of while Search the positon of minimum fitness value: *idx* ← *min-index*(*best_v*) Return *best_p*[*idx*]

## 3. One-Dimension Convolutional Neural Network

The one-dimension convolutional neural network (1D-CNN) is a classic deep feedback neural network. By which, the internal features in the input data can be extracted based on the convolution operation. Different from the traditional 2D-CNN model, the convolution operation on the input data are only performed on one dimension. Additionally, this makes the 1D-CNN model very suitable for processing one dimension series such as the time domain acceleration signal. By Ref. [34], a typical 1D-CNN model is constructed by the convolution layer, activation layer, pooling layer mainly, and the probability of classification is calculated by the classifier layer finally.

The convolution layer which contains a convolution kernel with learnable parameters is the core of 1D-CNN model. Additionally, the feature map is obtained by the convolution operation of convolution kernel with input data along the length. During the convolution operation, only a fragment of input data are processed by the kernel, and the parameters are shared with other kernels, which reduce the count of parameters and the difficulty of model training. The convolution operation is shown in Equation (14).
(14)$$y_{i}^{l}=w_{i}^{l}*y^{l-1}+b_{i}^{l}$$
where $y_{i}^{l}$ is the *i*-th feature in the output of *l*-th layer, $w_{i}^{l}$ represent the *i*-th weight matrix in *l*-th layer, b is the bias, $y^{l-1}$ is the output of last layer, and * represent the convolution operation.

After the processing of convolution layer, the obtained feature map is a linear mapping of input data and should be improved to discriminate the non-linear data. Therefore, the activation layer is adopted to add non-linear character into the feature maps. The refined linear unit (ReLU) is usually adopted for the activation function, which is shown in Equation (15).
(15)ReLUx=max0,x
where *x* is the input feature map.

The count of features in the feature map obtained by the convolution layer and activation layer are usually too big and increase the training difficulty of the model. To tackle this problem, the pooling layer is adopted to reduce the dimension of the feature map, and the input data can be represented by the other feature map with a smaller size. By the pooling operation, the length of input data for the next layer is reduced obviously, which is helpful to alleviate the over-fitting phenomenon in the training process. There are two kinds of pooling operations, average and maximum pooling. However, the maximum pooling is similar to the envelop demodulation in the field of traditional signal processing. Therefore, the maximum pooling is selected for the 1D-CNN model to process acceleration signal of rolling bearing, as is shown in Equation (16).
(16)$$y_{}^{l}\left[i\right]=\underset{\left(i-1\right)k+1\leq n\leq ik}{\max}\left\{y^{l-1}\left[n\right]\right\}$$
where *y^l^*[*i*] represents the output of *i*-th pooling region, and *k* is the length of the kernel of pooling layer.

Processed by several groups of convolution activation pooling layers, the feature map of the raw input data will be feed to the classifier of 1D-CNN model. The features are processed by the fully connected layer firstly, and next, the probability of classification is calculated by the softmax function. The training of model is to repeat the steps above iteratively; after that, the cross-entropy loss function is adopted to evaluate the model. According to the value of loss function, the parameters of model are updated iteratively until the end of training.

## 4. The Proposed Method

On the basis of the theories mentioned above, an integrated approach for fault diagnosis of rolling bearing, based on the BAS-SWD and 1D-CNN models, is proposed in this study. The main procedure of the proposed method is shown in Figure 2, the key steps are as follows.

Step 1. The acceleration signals of horizontal and vertical channels at the bearing pedestal where the fault bearing located are collected from the test rig. By replacing bearings with different faults, groups of the multi-channel signals with sufficient number are obtained.

Step 2. According to the speed of motor and the sampling frequency of signals, the collected signals are divided into fragments with fixed length. After that, the datasets with enough data samples are acquired. Then, the BAS-SWD algorithm is employed to extract the top three OCs with the highest value of envelope spectral kurtosis from the fragments. Finally, the datasets are constructed by these OCs, and divided into training samples, validate samples and test samples after well-labeled.

Step 3. Reference to the famous VGG-16 model, the 1D-CNN model is established. With this model, the features of each channel of signal are extracted by the convolutional-pooling layers, and the fused by the fusion layer. Finally, the input samples are classified by the softmax classifier.

Step 4. The 1D-CNN model is trained and tested by the acquired datasets. After that, the well-trained model is applied to realize the end-to-end fault diagnosis of rolling bearings.

## 5. Experimental Validations

### 5.1. Data Information

In this study, two datasets form the bearing datacenter in Case Western Reserve University (CWRU) and the wind turbine drivetrain diagnostics simulator (WTDS) in the author’s laboratory are adopted to verify the proposed fault diagnosis method for rolling bearing.

In the CWRU’s dataset, the acceleration signals under 2 HP of the bearings at the drive-end of the motor with a sampling frequency of 48 kHz are used as the raw data. The type of bearing installed at the drive-end of motor is 6205-2RS, whose manufacture is SKF. According to the requirement of the input layer of 1D-CNN model, the raw signals are resampled by a sampling frequency of 24 kHz and segmented into fragments with fixed length. The length of fragments is the quantity of sampling points during two revolves of the motor, the value in this paper is 1645 (24,000 × 60 × 2/1750).

There are some artificial wearing points on the inner race (IR), outer race (OR), and rolling elements (RE) with different sizes. Additionally, from the raw dataset, a total of seven types of signals are selected for the classification task. Therefore, the experimental dataset contains 7700 samples for the training, validating, and testing of the model; the details are shown in Table 1. 

In the other dataset, the acceleration signals collected from the WTDS test rig is with a sampling frequency of 20,480 Hz, the drive speed of motor is 1500 rpm, and the load is about 45 Nm. According to the WTDS test rig, the type of fault bearing is ER-16K. Additionally, there are two channels of acceleration signals in the dataset, which correspond to the horizontal and vertical installation of sensors. There is also one channel of tachometer pulses signal in the dataset. Similar to the CWRU’s dataset, the raw signals are segmented into fragments. Due to the bearing that is installed on the second shaft of the parallel gearbox and transmission ratio between the first and second shaft (36/90), the length should be 2048 (5 revolves on the motor).

The human-made faults are located respectively on the inner race, outer race, and rolling elements. Therefore, a total of four types of signals, with one normal and three faults, are contained in the dataset. The details can be seen in Table 2. 

The time domain waveforms of the signals in WTDS’s dataset are shown in Figure 3. By the figure, the differences among the different faults are not very clear and cannot be diagnosed easily. Therefore, the weak features in the raw signals need to be extracted for the classification.

### 5.2. Preprocessing of Signals

The proposed BAS-SWD algorithm is adopted to highlight the weak features in the noisy raw signals in this paper, so as to increase the accuracy of the 1D-CNN model’s classification result. 

Take the signal segment with the rolling element fault as an example, which is shown in Figure 3d. To obtain the value range of the SWD model’s parameters (*P_ωth_*, *T_th_*), many decomposition experiments were carried out. According to the results, when the value of *P_ωth_* is lower than 0.05, more OCs will be obtained; however, the energy of envelope spectrum of each OC is very low and no new frequency components exist in the OCs, so the minimum value of *P_ωth_* is set to 0.05. Meanwhile, when the value of *T_th_* is greater than 0.35, the calculation time will increase significantly, even falling into an infinite loop; so, the maximum value of *T_th_* is set as 0.35. The other initial parameters of BAS are listed in Table 3, and the fitness function is based on Equation (13).

Due to the advantages of fast convergence and strong global optimization ability in BAS algorithm, the value of fitness function converges stably after 12 iterations of calculation, as is shown in Figure 4. At this point, the value of fitness function is 0.0027, and the parameter pair of SWD is (0.15, 0.1).

To verify the advantages of BAS-SWD algorithm, the decomposition result is compared with the original SWD (*Pω_th_* = 0.1, *T_th_* = 0.1). For the selected signal segment, the rotational frequency of motor is 25 Hz (1500 rpm), and the rotational frequency of the second shaft connected to the bearing is 10 Hz (25 × 36/90), finally, the fault frequency is *f_re_* = 23.3 Hz (2.33 × 10). The waveforms and corresponding envelop spectra of the original SWD are shown in Figure 5. There are 4 OCs and 1 residual component extracted from the original signal fragment. The envelop spectrum indicated that the principal modulating component is caused by the meshing of gears between the first and second shafts (*f_r_* = 25 Hz), the fault frequency of the bearing is too weak to be observed clearly. Meanwhile, with the well-chosen parameters, the BAS-SWD algorithm extract 2 OCs and 1 residual component from the raw signal in Figure 6. Similarly, the envelop spectrum is used as the tool to analyze the result. Though the fault frequency is very weak, it can be observed in the 2nd and residual component (Figure 6b).

Based on the above analysis, the BAS-SWD’s ability of extracting weak features from noisy signals is improved. Additionally, the SNR of OC is higher than the raw signals. Therefore, the OCs are more suitable for training the deep learning model than the raw signal itself.

### 5.3. Construction of 1D-CNN Model

In this study, the 1D-CNN model based on VGG-16 is adopted to classify the rolling bearing’s acceleration signals. In VGG-16, the larger convolutional kernels are replaced by the cascading smaller ones (under the same sensory field), feature extraction ability is enhanced, and number of parameters is reduced [36]. Meanwhile, to further improve the stability and accuracy of the 1D-CNN model, some improvements are introduced into the model, which are shown in Figure 2.

After the preprocessing of signal, the top three OCs, with the highest envelope spectral kurtosis and correlations of each channel, are the input data of 1D-CNN model. Therefore, the input channel of 1D-CNN model is three. Meanwhile, there are two independent convolutional blocks with similar parameters in the convolutional part of the model, which corresponds to the signals from different channels of sensors. In the classification layer, the outputs of each convolutional layer are fused in the fusion layer to realize the feature level fusion firstly. Based on this, the weak feature in the raw signals could be enhanced. Next, the fusion features are processed by the combinations of the fully connected and dropout layers. Usually, the size of the feature map produced by flattened or fusion layer is much larger than the size of the fully connected layer’s output. Additionally, this increase the risk of over-fitting in the training progress, especially when the count of the training samples is not very large. According to the VGG-16 model, multiple fully connected layers are adopted in the proposed model. Additionally, the dropout layer is also adopted to add the non-linear characters to the fully connected layer, so as to improve the generalization ability. At last, the predicted classification of input signal is calculated by the cross-entropy loss function. In order to improve the ability of feature expression of the model, the stride value of max-pooling layer in each convolutional blocks, except the first one, is changed to 4 to extract more features. Based on the datasets collected from the WTDS test rig, the structure of the proposed 1D-CNN model is shown in Table 4. 

For different datasets, the parameters such as input size of FC1 and FC3 should be fine-turned to adjust the samples. The proposed model is built under the PyTorch 1.10.1 with no GPUs, and the version of python is 3.8. The OS of the host is Ubuntu 18.04 LTS, and the processor of the host is Intel Core^TM^ I5-4590 @ 3.3 GHz, and the memory is 16 GB.

In the training progress, the adopted back propagation algorithm is mini-batch gradicet descent, and the size of mini-batch is 20. Meanwhile, the adopted optimizer is Adam, and the initial learning rate is 0.001.

### 5.4. Case Study I

In this sector, the datasets from WTDS test rig were adopted to evaluate the proposed model. According to Table 2, there are four types of fault status of rolling bearing and 1100 samples for each status, with the length of 2048 acceleration values. The dataset is divided into training, validating, and testing sets by the ratio of 8:2:1. Additionally, the maximum number of iterations is set to 35.

The training set is normalized by min-max strategy, and then preprocessed by the BAS-SWD algorithm. After that the multi-channels samples of different sensors are fed into the model to train it. The loss value produced by the loss function represent the training progress of the model. The loss validation curves are shown in Figure 7. It can be concluded that the model has a fast convergence speed; the validate accuracy exceed 99% after about 12 iterations. That indicated the proposed model has been trained to the optimal level and has a good ability to classify the fault signals of rolling bearing.

Based on the trained model, the testing set is adopted to evaluate the classification performance, and the result is shown in Figure 8. The average classification accuracy of the proposed 1D-CNN model is 100% after many classification experiments.

Inside the model, the features were extracted from different channels of sensors by two independent convolutional blocks and fused in the fusion layer. To verify the advantages of feature fusion strategy, the classification result is compared to the result without feature fusion. Additionally, as is shown in Figure 9, the classification result with feature fusion is higher than the single channel signals, and it can also be seen that the classification result of the single channel V is better than channel H. This may be caused by the installation position of acceleration sensor.

The comparative experiments are carried out between the propose model and other models, such as the 1D-CNN model with raw signals, as well as the LSTM model with raw and decomposed signals. The results are shown in Table 5. The hidden units of LSTM in the comparative experiments were set to 32, the initial learning rate was 0.05, and the max iteration was set to 50. The LSTM model has some advantages in the learning time series and achieved good results in NLP fields. Due to the excellent feature extraction ability, the classification accuracy of 1D-CNN model is slightly higher than LSTM model, in both the raw and decomposed signals. For raw signals, the accuracy of the proposed model is 100% too, but the training time of the model of same structure is almost twice as the decomposed signals.

### 5.5. Case Study II

In this sector, the CWRU’s dataset, which has been adopted by many other researches, is used as the input data of 1D-CNN model. Different from the WTDS’s dataset, there is only one channel in the samples. Therefore, one of the two convolutional blocks, as well as the fusion layer, in the proposed 1D-CNN model are removed, and the input size is also changed to adjust the sample in the CWRU’s dataset. The other parameters of the model are same as the model for WTDS’s dataset.

After the training progress, the loss validation curves are shown in Figure 10. Due to the change in the structure of the model, the training time consumption is reduced. However, without the feature fusion, the convergence speed is slower. The loss validation curves arrive to a stable status when the 25th iteration terminated.

The confuse matrix of the classification result is shown in Figure 11. The average accuracy of classification is 100%. Compared to the classification result in Section 5.4, even the structure of model is concise, the accuracy of classification is same. The reason may be that the rolling bearing in CWRU’s test rig is on the shaft connected to the motor directly. On the other hand, the rolling bearing in WTDS’s test rig is on the second shaft, and the shaft is connected to the motor through a pair of gears. This transmission structure introduced another strong modulation component in the acceleration signals and made it more difficult to diagnose the fault of the rolling bearing.

Similar to the Section 5.4, the comparative experiments among different models are carried out to verify the stability of the proposed model. The result is shown in Table 6. The 1D-CNN model performed well on the raw and decomposed signals, and the only difference between them is that the model training progress based on the decomposed signals required less training time, and the convergence speed is faster than the other one. This indicate that the feature extraction ability of 1D-CNN model is stronger than other machine learning models.

## 6. Conclusions

In this study, an integrated intelligent method of fault diagnosis for rolling bearing was proposed, which was based on BAS-SWD and 1D-CNN. 

Within the proposed method, the complex, non-stationary acceleration signals collected from test rig were trimmed and normalized by min-max strategy firstly. Afterwards, to overcome the problem of hard to choose suitable parameter pair (*P_ωth_*, *T_th_*) of SWD, the BAS algorithm was adopted to obtain the optimal parameters of SWD, and the preprocessed acceleration signals were decomposed into several OCs by BAS-SWD adaptively. Finally, the 1D-CNN model with feature fusion layer was constructed for classification of the fault signals. The comparative experiments, based on different datasets, were carried out to evaluate the proposed method, and some conclusions can be drawn, as follows.

The SWD algorithm is suitable for preprocessing the non-stationary acceleration signals of rolling bearings, the weak modulation component caused by bearing’s fault can be extracted from the raw signals. Additionally, combined with the BAS optimal algorithm, the proposed BAS-SWD is more adaptive, effective, and efficient than the original SWD algorithm.

The 1D-CNN model has advantages in feature extraction. Compared with other deep learning models, the raw signals can be adopted as input data directly and without additional feature extraction progress. However, the feature extraction is very important for the improvement of accuracy of deep learning model. Therefore, the proposed 1D-CNN model is more suitable for the classification of the rolling bearing’s fault signals.

For the acceleration signals with complex modulation components and weak features, the feature fusion of multi-channels of signals can improve the performance of the deep learning model. Based on feature fusion, more features are extracted from the input data, which can improve the generalization ability of model.

The result of the comparative experiments, based on different datasets, indicate that the proposed method is effective, accurate, and stable. Additionally, it can provide an end-to-end alternative solution for rolling bearing’s fault diagnosis.

Future research will expand the application field of this method, such as the mixed faults of the gear and bearing or other rotation machines.

## Figures and Tables

**Figure 1 entropy-24-00573-f001:**
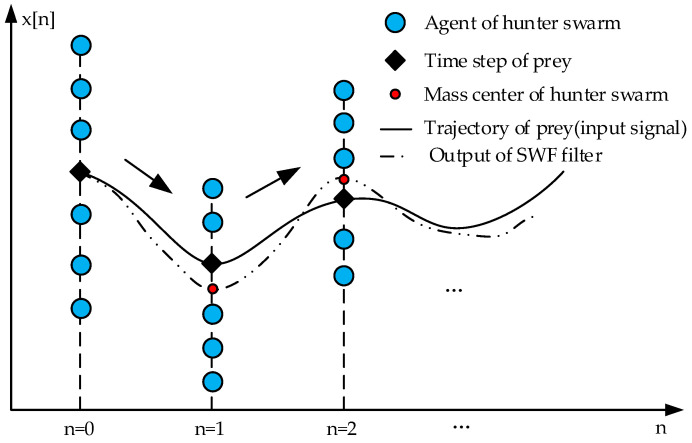
Procedure of filtering input signal by SWF model.

**Figure 2 entropy-24-00573-f002:**
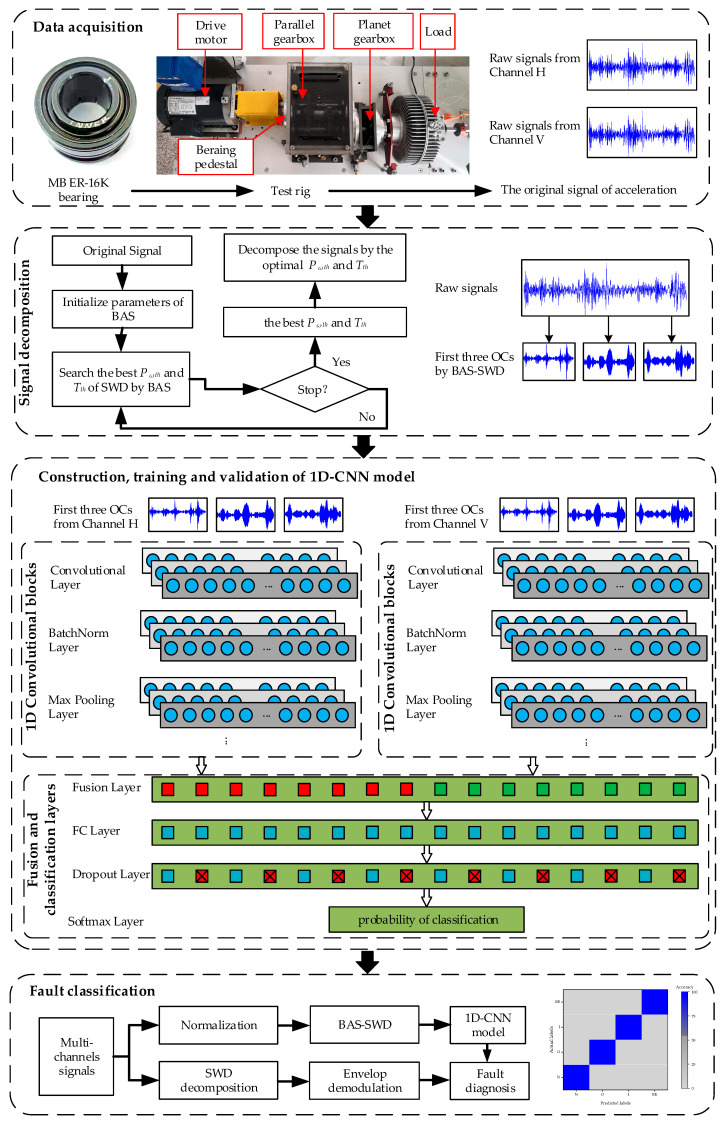
The proposed fault diagnosis method of rolling bearing.

**Figure 3 entropy-24-00573-f003:**
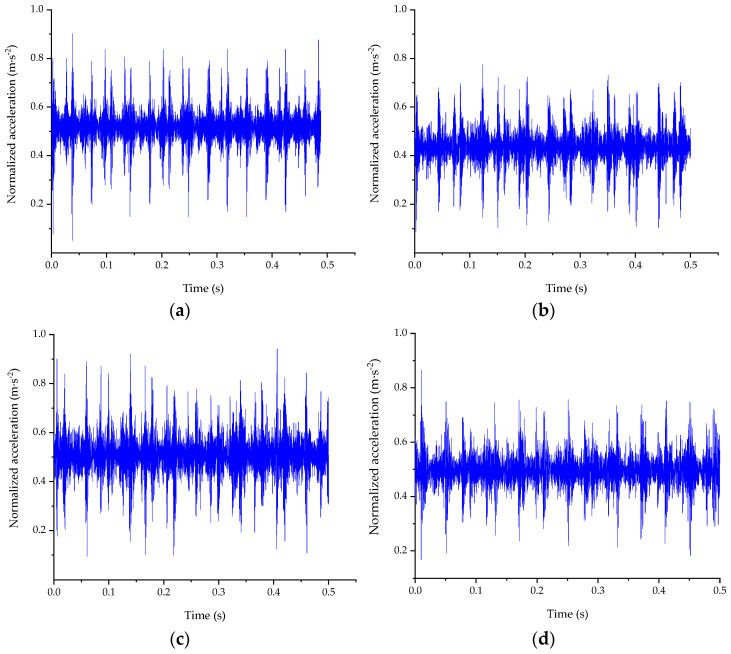
Time domain waveforms of WTDS dataset. (**a**) Normal bearing; (**b**) outer race fault; (**c**) inner race fault; (**d**) rolling element fault.

**Figure 4 entropy-24-00573-f004:**
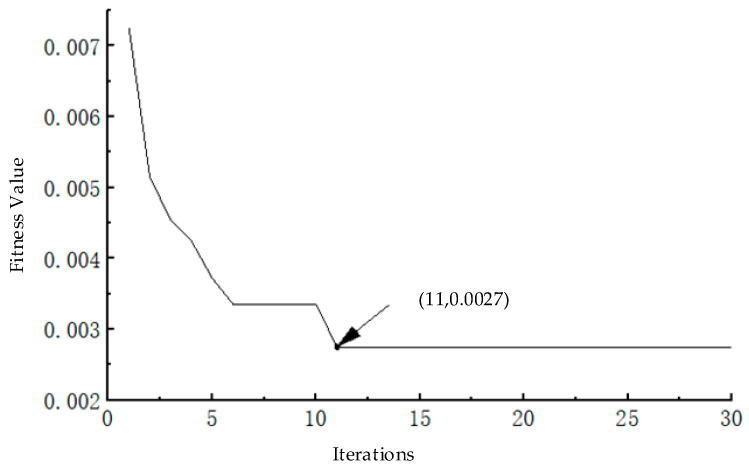
Convergence curve of BAS-SWD.

**Figure 5 entropy-24-00573-f005:**
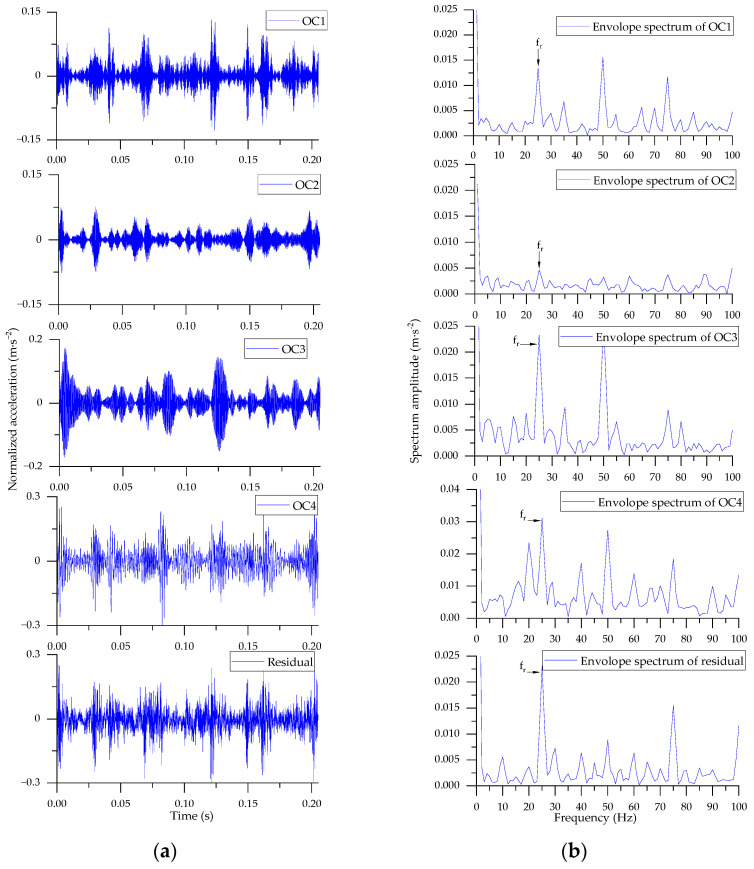
Decomposition result of SWD. (**a**) Waveforms of OCs; (**b**) Envelope spectrum of OCs.

**Figure 6 entropy-24-00573-f006:**
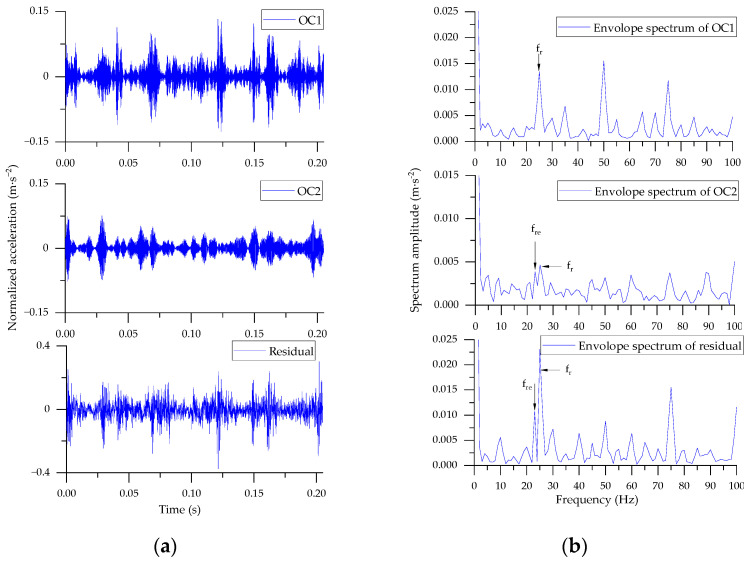
Decomposition result of BAS-SWD. (**a**) Waveforms of OCs; (**b**) Envelope spectrum of OCs.

**Figure 7 entropy-24-00573-f007:**
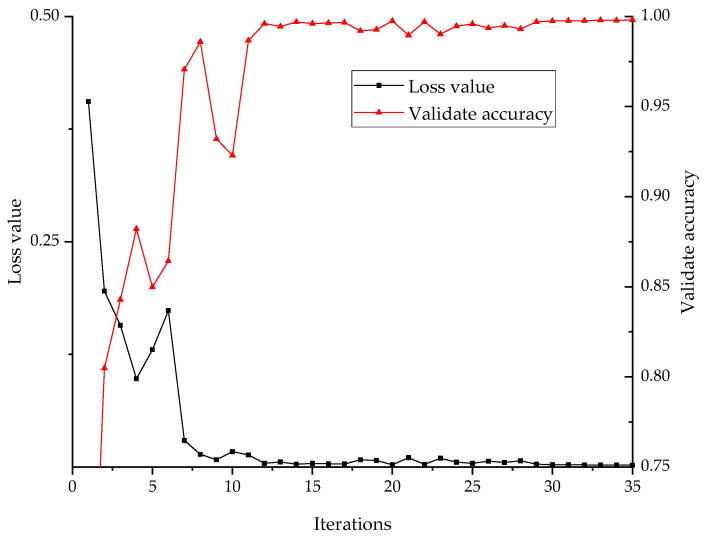
Loss validation curves of the proposed 1D-CNN model based on WTDS’s dataset.

**Figure 8 entropy-24-00573-f008:**
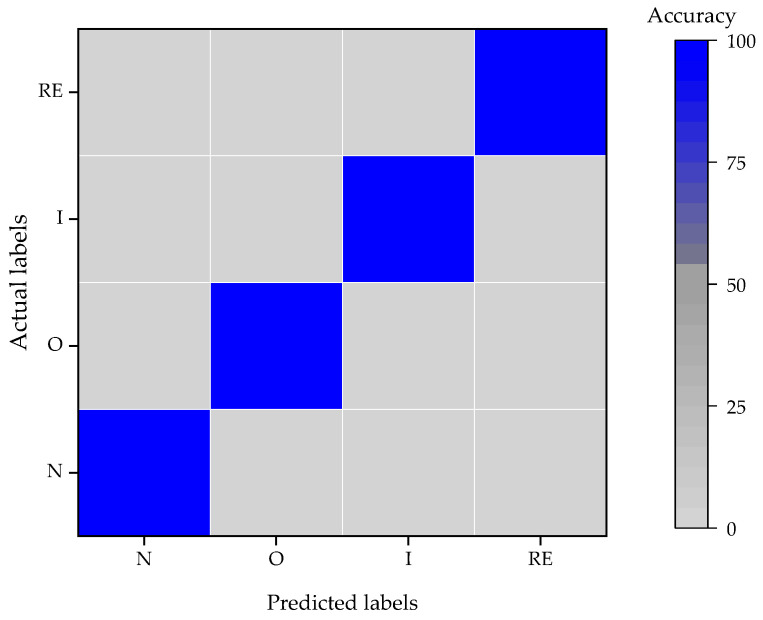
Classification result of WTDS dataset.

**Figure 9 entropy-24-00573-f009:**
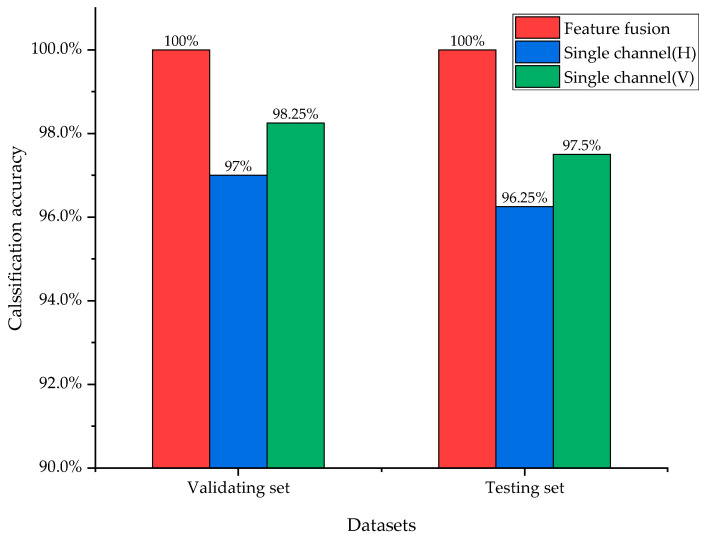
Classification results of feature fusion and no fusion.

**Figure 10 entropy-24-00573-f010:**
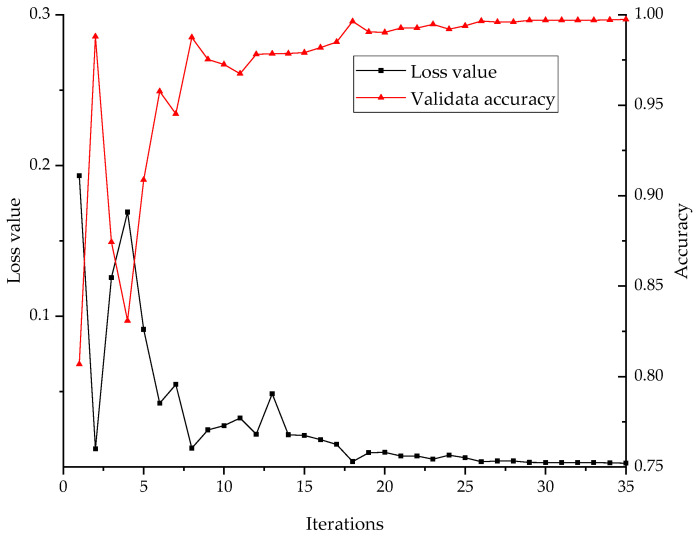
Loss-validation curves of 1D-CNN model based on CWRU’s dataset.

**Figure 11 entropy-24-00573-f011:**
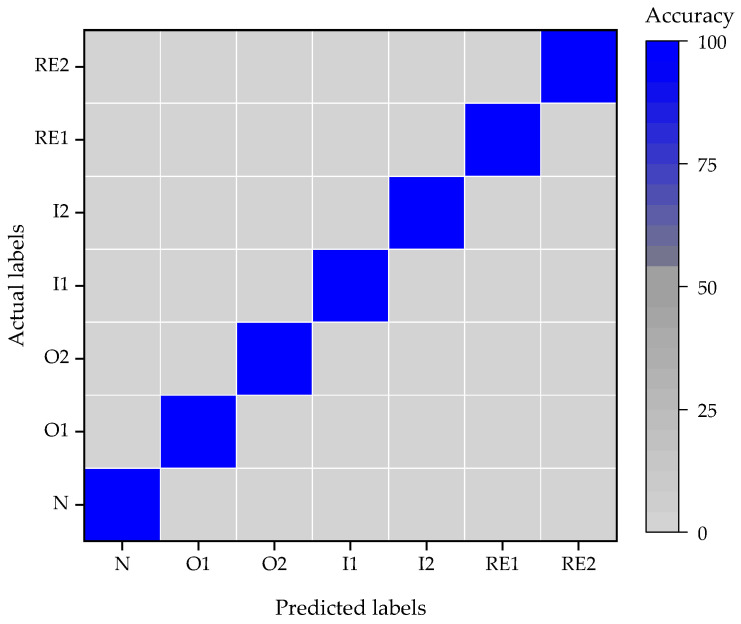
Classification result of CWRU’s dataset.

**Table 1 entropy-24-00573-t001:** Details of the experimental dataset selected from CWRU’s data.

Label	Fault Information	Length of Sample	Count of Samples	Training Set	Validation Set	Testing Set
N	Normal	1645	7700	5600	1400	700
O1	Outer race 0.007″					
O2	Outer race 0.014″					
I1	Inner race 0.007″					
I2	Inner race 0.014″					
RE1	Rolling Elements 0.007″					
RE2	Rolling elements 0.014″					

**Table 2 entropy-24-00573-t002:** Details of the experimental dataset collected from the author’s laboratory.

Label	Fault Information	Length of Sample	Count of Samples	Training Set	Validation Set	Testing Set
N	Normal	2048	4400	3200	800	400
O	Outer race					
I	Inner race					
RE	Rolling elements					

**Table 3 entropy-24-00573-t003:** Parameters of BAS.

*L*	*D* _0_	*δ* _0_	*n*	*l_b_*	*u_b_*
30	0.01	0.95	2	[0.05, 0.01]	[0.99, 0.35]

**Table 4 entropy-24-00573-t004:** Structure of the proposed 1D-CNN.

Layer	Kernel Count and Size	Stride	Padding	Input Size	Output Size
Conv 1	64-1 × 3	1	1	Batch × 3 × 2048	Batch × 64 × 2048
BN/AC 1	-	-	-	Batch × 64 × 2048	Batch × 64 × 2048
Conv 2	64-1 × 3	1	1	Batch × 64 × 2048	Batch × 64 × 2048
BN/AC 2	-	-	-	Batch × 64 × 2048	Batch × 64 × 2048
Pooling 1	64-1 × 2	2	0	Batch × 64 × 2048	Batch × 64 × 1024
Conv 3	128-1 × 3	1	1	Batch × 64 × 1024	Batch × 128 × 1024
BN/AC 3	-	-	-	Batch × 128 × 1024	Batch × 128 × 1024
Conv 4	128-1 × 3	1	1	Batch × 128 × 1024	Batch × 128 × 1024
BN/AC 4	-	-	-	Batch × 128 × 1024	Batch × 128 × 1024
Pooling 2	128-1 × 4	4	0	Batch × 128 × 1024	Batch × 128 × 256
Conv 5	256-1 × 3	1	1	Batch × 128 × 256	Batch × 256 × 256
BN/AC 5	-	-	-	Batch × 256 × 256	Batch × 256 × 256
Conv 6	256-1 × 3	1	1	Batch × 256 × 256	Batch × 256 × 256
BN/AC 6	-	-	-	Batch × 256 × 256	Batch × 256 × 256
Conv 7	256-1 × 3	1	1	Batch × 256 × 256	Batch × 256 × 256
BN/AC 7	-	-	-	Batch × 256 × 256	Batch × 256 × 256
Pooling 3	256-1 × 4	4	0	Batch × 256 × 256	Batch × 256 × 64
Conv 8	512-1 × 3	1	1	Batch × 256 × 64	Batch × 512 × 64
BN/AC 8	-	-	-	Batch × 512 × 64	Batch × 512 × 64
Conv 9	512-1 × 3	1	1	Batch × 512 × 64	Batch × 512 × 64
BN/AC 9	-	-	-	Batch × 512 × 64	Batch × 512 × 64
Conv 10	512-1 × 3	1	1	Batch × 512 × 64	Batch × 512 × 64
BN/AC 10	-	-	-	Batch × 512 × 64	Batch × 512 × 64
Pooling 4	512-1 × 4	4	0	Batch × 512 × 64	Batch × 512 × 16
Conv 11	512-1 × 3	1	1	Batch × 512 × 16	Batch × 512 × 16
BN/AC 11	-	-	-	Batch × 512 × 16	Batch × 512 × 16
Conv 12	512-1 × 3	1	1	Batch × 512 × 16	Batch × 512 × 16
BN/AC 12	-	-	-	Batch × 512 × 16	Batch × 512 × 16
Conv 13	512-1 × 3	1	1	Batch × 512 × 16	Batch × 512 × 16
BN/AC 13	-	-	-	Batch × 512 × 16	Batch × 512 × 16
Pooling 5	512-1 × 4	4	0	Batch × 512 × 16	Batch × 512 × 4
Flatten 1	-	-	-	Batch × 512 × 4	Batch × 1 × 2048
Fusion 1				Batch × 1 × 2048	Batch × 1 × 4096
FC 1	-	-	-	Batch × 1 × 4096	Batch × 1 × 512
Dropout 1	Dropout rate 0.5			Batch × 1 × 512	Batch × 1 × 512
FC 2	-	-	-	Batch × 1 × 512	Batch × 1 × 512
Dropout 2	Dropout rate 0.5			Batch × 1 × 512	Batch × 1 × 512
FC 3	-	-	-	Batch × 1 × 512	Batch × 1 × 4

**Table 5 entropy-24-00573-t005:** Classifications of different models.

Model Names	Accuracy on Testing Sets
Proposed model with feature fusion	100%
Proposed model with raw signals with feature fusion	100%
Proposed model with decomposed signals from channel V	98.25%
Proposed model with raw signals from channel V	98.50%
LSTM model with decomposed signals from channel V	93.75%
LSTM model with raw signals from channel V	92.5%

**Table 6 entropy-24-00573-t006:** Classifications of different models, based on the CWRU’s dataset.

Model Names	Accuracy on Testing Sets
Proposed model with decomposed signals	100%
Proposed model with raw signals	100%
LSTM model with decomposed signals	95%
LSTM model with raw signals	94.25%
TSFFCNN Ref. [27]	97%

## Data Availability

The public dataset from CWRU can be found at https://engineering.case.edu/bearingdatacenter (accessed on 1 March 2022). The signals collected form WTDS test rig in the author’s laboratory are owned by the author’s institution and not allowed to be used in the public way.

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
