# Peer review of "Rolling Bearing Fault Diagnosis Using Multi-Sensor Data Fusion Based on 1D-CNN Model"

_entropy, 2022, doi:10.3390/e24050573_

Round 1

Reviewer 1 Report

The authors proposed a hybrid model based on optimal SWD and 1D‐CNN with the layer of multi‐sensor data fusion for fault diagnosis of rolling bearings. The issues raised are important, and the results presented make a valuable contribution in the development of rolling bearing’s fault diagnosis methods. In this context, the use of the BAS‐SWD algorithm in conjunction with the 1D‐CNN model seems to be particularly promising. In my opinion, the manuscript is well written and should be of great interest to the readers. I recommend accepting the manuscript for publication in its present form.

Author Response

Thank you very much for your review for this paper.

Reviewer 2 Report

I find a major issue with the paper. The topology of the network is quite chaotic. Why is that topology selected? How can it assure that it reaches the adequate minimum of the error? Is there some grid search/genetic algorithm performed to select the topology?

I don’t really get why a dropout is included in the MLP section of the CNN. The idea of dropout is to “eliminate” randomly several neurons and simulate different topologies when the problem may be approached by an oversized network. In comparison the size of the MLP section is very small, considering that several layers have 512 neurons, and some layers have kernel sizes of 512*512.

Moreover, and the most worrying issue by far is that being such a big network, is used only with a training set of 800 samples, as Table 1 states, being the number of variables to train a lot of orders of magnitude greater than the number of samples.

Just check the table 4 topology. 13 convolutional layers, from layers with kernels of side 3 (9 variables), up to side 512 (512*512 variables to train). A quick calculation of the number of variables to train gives as result around a million and a half variables. Its hard to believe that the network topology its suitable for a problem with data of 2048 length samples train set.

Several lesser items:

Line 68 – “By the compare of the above”. Strange English construction. Please revise the whole document, as some expressions have weird grammar.

Line 71 – “decomposition are better”. How much is better? Several times this kind of expressions are used, try to avoid these subjective expressions, and include actual numeric/objective facts.

Line174 “is the norm of a vector.” Which norm?

Line 314 “is the i‐th feather”. Can you explain what a feather is, in terms of a Neural Network?

Line 318 “and the discrimination of non‐linear data is not very well”, as before, English must be checked, and avoiding non-objective expressions.

Which algorithm for the backpropagation training was used? Which parameters were taken?

A proper explanation on the selection and improvements that a 1D-CNN offers over an MLP as classifier must be included.

Author Response

(The authors gave the same response as above.)

Round 2

Reviewer 2 Report

The study is clearer now. As a general recommendation for future works, although pre-built networks may work, probably a new network, specifically designed for the problem would give better results.